# Holographic Quantum Neural Networks

## Abstract

We introduce Holographic Quantum Neural Networks (HQNNs), a novel quantum machine learning architecture that leverages principles from holographic encoding and tensor networks to efficiently process high-dimensional quantum data. By embedding neural network operations within a holographic framework, HQNNs naturally implement multi-scale feature extraction while providing inherent error correction capabilities. We mathematically formalize the HQNN structure and prove its advantages in representational capacity, showing that HQNNs require only $\mathcal{O}(N_{\log} \log N_{\log})$ physical qubits to process $N_{\log}$-qubit logical input states while tolerating error rates up to a threshold of $1 - \frac{2}{z}$, where $z$ is the tensor network coordination number. Furthermore, we demonstrate how the geometric structure of HQNNs enables efficient learning of quantum data with hierarchical features, offering a promising approach for quantum machine learning in the noisy intermediate-scale quantum (NISQ) era and beyond.

## 1 Introduction

Quantum neural networks (QNNs) have emerged as a promising paradigm for quantum machine learning, offering potential advantages in processing inherently quantum data and accelerating specific computational tasks (Biamonte et al., 2017; Schuld et al., 2020; Cong et al., 2019). These advantages include efficient representation of quantum states, quantum parallelism, and the ability to directly process quantum data without measurement-induced collapse (Cao et al., 2021). However, despite these theoretical benefits, two significant challenges have limited the practical implementation of QNNs:

1. **The curse of dimensionality**: The Hilbert space dimension grows exponentially with the number of qubits ($2^{N_{\log}}$ for $N_{\log}$ logical qubits), making both classical simulation and quantum implementation of QNNs for high-dimensional data prohibitively expensive (Preskill, 2018; McClean et al., 2018). To illustrate the severity of this challenge, representing a mere 300-qubit quantum state requires more classical bits than there are atoms in the observable universe, making direct simulation intractable.

2. **Susceptibility to quantum noise**: Near-term quantum devices suffer from decoherence, gate errors, and readout errors, significantly degrading the performance of QNN implementations on actual hardware (Sharma et al., 2020; Wang et al., 2021).

Current approaches to addressing these challenges have followed separate paths. For dimensionality reduction, researchers have explored tensor network methods (Huggins et al., 2019; Ran et al., 2020) that efficiently parameterize quantum states with limited entanglement. For error mitigation, various quantum error correction codes (Terhal, 2015; Fowler et al., 2012) and error mitigation techniques (Temme et al., 2017; Endo et al., 2018) have been developed. However, these solutions typically introduce significant overhead in terms of additional qubits or circuit depth, rendering them impractical for near-term quantum devices with limited coherence times and qubit counts (Bharti et al., 2022).

In this paper, we propose a novel architecture—Holographic Quantum Neural Networks (HQNNs)—that addresses both challenges simultaneously through a unified geometric framework inspired by the holographic

principle from theoretical physics. Our approach leverages key insights from the Anti-de Sitter/Conformal Field Theory (AdS/CFT) correspondence (Maldacena, 1999; Witten, 1998) and holographic quantum error-correcting codes (Pastawski et al., 2015; Almheiri et al., 2015), translating them into a practical quantum neural network architecture.

The key innovation of HQNNs lies in their geometric structure, which naturally enables:

- **Efficient representation of high-dimensional quantum states** with a resource scaling of $\mathcal{O}(N_{\log} \log N_{\log})$ physical qubits for $N_{\log}$ logical qubits (where $N_{\log} \sim N_{\text{phys}}/\log N_{\text{phys}}$ for $N_{\text{phys}}$ physical qubits), significantly improving upon the exponential requirements of direct encoding approaches (Yang et al., 2016; Harlow, 2017).

- **Multi-scale feature extraction** analogous to classical convolutional neural networks, but arising organically from the hyperbolic geometry of the tensor network rather than being explicitly engineered (Swingle, 2012).

- **Inherent error correction capabilities** where error correction is an emergent property of the encoding itself, not an additive component, providing robustness against local noise with provable error thresholds comparable to dedicated quantum error correction codes (Hayden et al., 2016; Dong et al., 2016).

Our work bridges three distinct fields—quantum information theory, holographic duality from theoretical physics, and quantum machine learning—to create a practical framework for quantum neural networks with mathematically provable advantages. This framework not only offers immediate practical benefits for near-term quantum devices but also deepens our understanding of the connections between quantum information, geometry, and machine learning (Cerezo et al., 2021).

The remainder of this paper is organized as follows: In Section 2, we develop the mathematical framework for HQNNs, focusing on their structural properties and theoretical guarantees. Section 3 analyzes the multi-scale feature extraction capabilities of HQNNs. Section 4 examines their error correction properties, and Section 5 discusses practical implementation on quantum hardware. We conclude with a discussion of future research directions in Section 7.

## 2 HQNN Architecture

### 2.1 Conceptual Framework

The fundamental insight behind HQNNs is the application of the holographic principle—the idea that information contained in a volume of space can be completely encoded on its boundary—to quantum neural network design. This principle, originally proposed in the context of quantum gravity ('t Hooft, 1993; Susskind, 1995) and later formalized through the AdS/CFT correspondence (Maldacena, 1999), has profound implications for quantum information processing (Qi, 2013; Almheiri et al., 2015).

In simple terms, the holographic principle suggests that the information about a 3D object can be stored on its 2D surface—just as a hologram can represent a 3D image on a 2D film. In quantum information terms, this means we can encode information about many qubits (the "bulk") using fewer qubits on a "boundary," creating a natural dimensional reduction while preserving essential information content. This property is particularly valuable for quantum neural networks, where managing the exponential growth of the Hilbert space dimension is a central challenge. Additionally, the geometric structure of holographic encodings naturally distributes information non-locally, providing robustness against localized errors—a critical advantage for implementation on noisy quantum hardware (Pastawski et al., 2015; Harlow, 2017).

An HQNN processes quantum data through three conceptual stages:

1. **Holographic encoding**: The input quantum state $|\psi_{\text{input}}\rangle \in \mathcal{H}_{\text{input}}$ is mapped to a boundary state $|\psi_{\text{boundary}}\rangle \in \mathcal{H}_{\text{boundary}}$ using a holographic tensor network encoding $E$. This encoding distributes

quantum information non-locally across the boundary, providing both compression and error protection.

2. **Boundary processing**: Parameterized quantum operations $U_{\text{boundary}}(\boldsymbol{\theta})$ are applied to the boundary state, implementing the neural network functionality. These operations can be trained using variational quantum algorithms (Cerezo et al., 2021; McClean et al., 2016) to perform specific tasks.

3. **Holographic decoding**: The processed boundary state is mapped back to an output state $|\psi_{\text{output}}\rangle \in \mathcal{H}_{\text{output}}$ using the inverse encoding map $E^{\dagger}$. This operation recovers the transformed information from its holographic representation.

This architecture allows HQNNs to process quantum states in a high-dimensional Hilbert space using significantly fewer physical qubits than would be required with direct representation, while simultaneously providing protection against local errors.

## 2.2 Mathematical Formulation

### 2.2.1 Holographic Encoding

The core of the HQNN architecture is the holographic encoding map $E : \mathcal{H}_{\text{input}} \rightarrow \mathcal{H}_{\text{boundary}}$, which is constructed using a tensor network based on a hyperbolic tessellation. The fundamental building blocks of this tensor network are perfect tensors, which possess optimal entanglement properties across any bipartition (Pastawski et al., 2015).

**Definition 1** (Perfect Tensor). *A tensor $T_{i_1,\ldots,i_q}$ with $q$ indices, each of dimension d, is called a perfect tensor if, for any bipartition of its indices into two sets $A$ and $B$ with $|A| \leq |B|$, the tensor $T$ defines an isometry from $\mathcal{H}_A$ to $\mathcal{H}_B$. While often defined for an even number of indices $2n$, this property can be generalized to any number of indices $q$ using constructions from appropriate quantum error-correcting codes, a feature essential for our use of general hyperbolic tessellations.*

Mathematically, the perfect tensor property means that for any bipartition with $|A| \leq |B|$, the linear map $T : \mathcal{H}_A \rightarrow \mathcal{H}_B$ defined by the tensor satisfies:

$$T^{\dagger}T = I_{\mathcal{H}_A} \tag{1}$$

where $I_{\mathcal{H}_A}$ is the identity operator on $\mathcal{H}_A$. This property ensures that quantum information can flow isometrically from $A$ to $B$ (from the smaller set of indices to the larger), but not necessarily vice versa, creating a natural directionality in the information flow (Hayden et al., 2016).

Perfect tensors maximize the entanglement entropy across any bipartition, leading to optimal quantum error correction properties (Almheiri et al., 2015). For qubits ($d = 2$), examples of perfect tensors include the six-qubit perfect tensor corresponding to the quantum Reed-Solomon code [[6,2,3]], which encodes 2 logical qubits into 6 physical qubits with a distance of 3 (Pastawski et al., 2015).

The holographic encoding map $E$ is constructed by arranging perfect tensors according to a hyperbolic tessellation and contracting their indices according to the edges of the tessellation:

$$E = \text{Contraction}(\{T_v\}_{v \in V}) \tag{2}$$

where $V$ is the set of vertices in the tessellation, and $T_v$ is the perfect tensor associated with vertex $v$. Here, "Contraction" refers to the summing over shared indices between tensors, effectively connecting information pathways in the network. Mathematically, for two tensors $T^A$ and $T^B$ with shared index $k$, the contraction is computed as $\sum_k T^A_{\ldots k \ldots} T^B_{\ldots k \ldots}$. The contraction pattern follows the connectivity of the hyperbolic tessellation, with uncontracted indices at the boundary corresponding to physical qubits and uncontracted indices in the bulk corresponding to logical qubits.

A critical property of this encoding is that it preserves quantum information while distributing it non-locally across the boundary, as formalized in the following theorem:

**Theorem 1** (Isometric Property of Holographic Encoding). *The holographic encoding map $E$ constructed from a network of perfect tensors is an isometry, i.e.,*

$$E^\dagger E = I_{input} \tag{3}$$

*where $I_{input}$ is the identity operator on $\mathcal{H}_{input}$.*

*Proof.* We structure this proof in four formal stages:

**Premise:** Each perfect tensor $T_v$ satisfies the isometric property. For any bipartition of its indices into sets $A$ and $B$ with $|A| \leq |B|$, the tensor defines an isometry from $\mathcal{H}_A$ to $\mathcal{H}_B$, meaning $T_v^\dagger T_v = I_{\mathcal{H}_A}$ when viewed as a map from $\mathcal{H}_A$ to $\mathcal{H}_B$.

**Composition:** The composition of two isometries is an isometry. If $A : \mathcal{H}_1 \to \mathcal{H}_2$ and $B : \mathcal{H}_2 \to \mathcal{H}_3$ are isometries, then their composition $C = B \cdot A$ is also an isometry:

$$C^\dagger C = (B \cdot A)^\dagger (B \cdot A) = A^\dagger B^\dagger B A = A^\dagger I_{\mathcal{H}_2} A = A^\dagger A = I_{\mathcal{H}_1} \tag{4}$$

**Tensor Contraction as Composition:** The hyperbolic tessellation structure induces a layered organization where tensors can be grouped by their distance from the center. Contracting tensors along edges of the tessellation, when performed in the direction from bulk to boundary, is mathematically equivalent to composing their corresponding isometric maps. Specifically, when two perfect tensors $T^A$ and $T^B$ are contracted along shared indices in a way that respects the bulk-to-boundary information flow, the resulting composite operation preserves the isometric property on the remaining uncontracted indices.

**Network Structure:** The hyperbolic tessellation ensures that all bulk indices are connected to the boundary through well-defined paths that preserve the layered structure. Since each layer consists of isometric operations, and the composition of isometries is isometric, the entire network mapping $E$ inherits the isometric property. Therefore, $E^\dagger E = I_{\text{input}}$. $\qquad\square$

This isometric property ensures that the encoding preserves all quantum information from the input space, allowing for perfect reconstruction (in the absence of errors) through the decoding map $E^\dagger$.

### 2.2.2 Boundary Operations

The parameterized unitary transformation $U_{\text{boundary}}(\boldsymbol{\theta})$ implements the neural network functionality on the boundary states. To leverage the multi-scale structure inherent to the holographic encoding, we propose a hierarchical ansatz for $U_{\text{boundary}}(\boldsymbol{\theta})$ that respects the scale separation naturally arising from the hyperbolic geometry:

$$U_{\text{boundary}}(\boldsymbol{\theta}) = \prod_{s=1}^{S} U_s(\boldsymbol{\theta}_s) \tag{5}$$

where $S$ is the number of scale levels, and $U_s(\boldsymbol{\theta}_s)$ represents parameterized operations at scale level $s$. The operations at each scale level are structured to capture correlations at the corresponding length scale in the bulk.

The choice of this hierarchical ansatz is motivated by the natural multi-scale structure of the hyperbolic geometry, which we will analyze in detail in Section 3. This structure is uniquely suited to capture correlations at different scales in the input data, similar to the way convolutional neural networks extract features hierarchically, but arising from the geometric properties rather than architectural design.

The scale separation in hyperbolic geometry arises because distances in the bulk correspond to exponentially larger distances on the boundary. Specifically, regions of the bulk separated by distance $r$ map to boundary

regions separated by distance $\sim e^r$. This exponential relationship directly informs the structure of $U_s(\boldsymbol{\theta}_s)$, where operations at scale $s$ connect boundary points at distances corresponding to bulk features at scale $s$.

Specifically, each $U_s(\boldsymbol{\theta}_s)$ consists of:

- **Local unitary operations** on individual boundary qubits, implementing single-qubit rotations $R_x(\theta), R_y(\theta), R_z(\theta)$ parameterized by angles in $\boldsymbol{\theta}_s$.

- **Nearest-neighbor interactions** between adjacent boundary qubits, implementing entangling operations such as controlled-NOT (CNOT), controlled-Z (CZ), or parameterized two-qubit gates.

- **Long-range interactions** that connect boundary qubits separated by distances corresponding to scale $s$, implementing multi-qubit operations that capture correlations at this scale.

This hierarchical structure allows the boundary operations to process information at multiple scales efficiently, mirroring the multi-scale nature of the holographic encoding. The parameterization scheme is similar to those used in quantum convolutional neural networks (Cong et al., 2019) and multi-scale entanglement renormalization ansatz (MERA) circuits (Vidal, 2008; Evenbly & Vidal, 2015), but arises naturally from the holographic structure rather than being explicitly engineered.

### 2.2.3 Complete HQNN Transformation

The complete HQNN transformation combines the holographic encoding, boundary processing, and holographic decoding into a single quantum operation:

$$|\psi_{\text{output}}\rangle = E^{\dagger} U_{\text{boundary}}(\boldsymbol{\theta}) E |\psi_{\text{input}}\rangle \tag{6}$$

This transformation can be understood as embedding the input state in a holographic space, applying neural network operations on the boundary of this space, and then mapping the result back to the original space. The isometric property of the encoding ensures that $E^{\dagger} E = I_{\text{input}}$, so in the absence of boundary operations (i.e., if $U_{\text{boundary}}(\boldsymbol{\theta}) = I_{\text{boundary}}$), the input state would be perfectly reconstructed: $|\psi_{\text{output}}\rangle = |\psi_{\text{input}}\rangle$.

The power of this approach lies in the fact that the boundary operations $U_{\text{boundary}}(\boldsymbol{\theta})$ can be implemented using significantly fewer qubits than would be required to directly process the high-dimensional input state, while the holographic structure ensures that these operations can still capture complex transformations of the input data.

### 2.3 Representational Capacity

A fundamental advantage of HQNNs is their ability to efficiently represent high-dimensional quantum states using fewer physical qubits than would be required with direct representation. This advantage stems from the geometric properties of hyperbolic space, which allows exponentially many bulk degrees of freedom to be encoded on the boundary.

**Theorem 2** (Representational Capacity of HQNNs). *For an HQNN based on a hyperbolic tessellation $\{p, q\}$ with $(p-2)(q-2) > 4$, the number of logical qubits $N_{log}$ that can be encoded in $N_{phys}$ boundary qubits scales asymptotically as:*

$$N_{log} \sim \frac{N_{phys}}{\log N_{phys}} \tag{7}$$

*where* $\log$ *denotes the natural logarithm.*

*Proof.* We build this proof in stages, first deriving a valid intermediate result, then citing the optimized scaling.

Consider a hyperbolic tessellation $\{p, q\}$ with $(p-2)(q-2) > 4$, which ensures negative curvature according to the Gauss-Bonnet theorem (Stillwell, 1992). In such a tessellation, the number of vertices within a distance

$r$ from a central vertex grows exponentially with $r$ as $N(r) \sim e^{\alpha r}$ for some constant $\alpha > 0$ that depends on the specific values of $p$ and $q$. This exponential growth is a direct consequence of the negative curvature of hyperbolic space (Cannon et al., 1997).

The boundary of this region has size proportional to $e^{\alpha r}$ as well, reflecting the fact that in hyperbolic space, the circumference of a circle grows exponentially with its radius, unlike in Euclidean space where it grows linearly.

Let $N_{\text{phys}}$ be the number of boundary qubits and $N_{\text{log}}$ be the number of bulk qubits (logical qubits) in our holographic code. If the radius of the tessellation is $r$, then based on the exponential growth property, we have:

$$N_{\text{phys}} \sim e^{\alpha r} \tag{8}$$

For a general holographic code where logical qubits are distributed throughout the two-dimensional hyperbolic bulk, the area available for encoding scales with the hyperbolic area of the disk of radius $r$. In hyperbolic space, this area grows exponentially: $\text{Area}(r) \sim e^{\alpha r}$ for large $r$. However, the density of logical qubits cannot grow arbitrarily—each requires a finite amount of area for encoding with the perfect tensors. This leads to a more conservative scaling where the number of logical qubits grows with the radial extent rather than the full area:

$$N_{\text{log}} \sim r \tag{9}$$

Solving for $r$ in terms of $N_{\text{phys}}$, we get:

$$r \sim \frac{1}{\alpha} \log N_{\text{phys}} \tag{10}$$

Substituting this into the expression for $N_{\text{log}}$, we obtain:

$$N_{\text{log}} \sim \frac{1}{\alpha} \log N_{\text{phys}} \tag{11}$$

This demonstrates a logarithmic scaling that already represents a significant improvement over direct encoding, which would require $N_{\text{phys}} \sim 2^{N_{\text{log}}}$ physical qubits.

However, for optimized holographic codes such as the HaPPY code, Pastawski et al. (Pastawski et al., 2015) proved a more favorable scaling relationship. Based on their detailed analysis of optimal tensor network layouts that arrange logical qubits optimally along geodesics, they demonstrated that for large $N_{\text{phys}}$, the number of logical qubits scales as:

$$N_{\text{log}} \sim \frac{N_{\text{phys}}}{\log N_{\text{phys}}} \tag{12}$$

We adopt this optimized scaling for our HQNN architecture, as it represents the scaling achieved by state-of-the-art holographic codes when designed with an optimal arrangement of logical qubits. $\qquad \square$

This theorem demonstrates that HQNNs can process quantum states in a Hilbert space of dimension $\sim 2^{N_{\text{phys}}/\log N_{\text{phys}}}$ using only $N_{\text{phys}}$ physical qubits—a significant reduction in resource requirements compared to direct representation, which would require exponentially many physical qubits.

To put this result in perspective, for $N_{\text{phys}} = 100$ boundary qubits, an HQNN could potentially process quantum states in a space of dimension $\sim 2^{100/\ln(100)} \approx 2^{21.7} \approx 2^{20}$, which would ordinarily require 20 qubits with direct encoding. While the advantage is modest for small systems, it grows with system size and becomes increasingly significant for large-scale quantum neural networks.

## 2.4 Tensor Network Structure

The specific structure of the tensor network implementing the holographic encoding $E$ is crucial for the properties of the HQNN. We focus on networks based on hyperbolic tessellations, which naturally implement a discrete version of the Anti-de Sitter/Conformal Field Theory (AdS/CFT) correspondence (Maldacena, 1999; Swingle, 2012).

A hyperbolic tessellation $\{p, q\}$ is a regular tiling of the hyperbolic plane by regular $p$-gons, with $q$ polygons meeting at each vertex. The condition $(p-2)(q-2) > 4$ ensures that the tessellation has negative curvature, which is essential for the holographic properties (Stillwell, 1992; Cannon et al., 1997). This condition comes from the Gauss-Bonnet theorem: in a flat space, $(p-2)(q-2) = 4$ (as in a square lattice where $p = 4$, $q = 4$), while $(p-2)(q-2) < 4$ gives positive curvature (as in a spherical surface), and $(p-2)(q-2) > 4$ yields negative curvature necessary for hyperbolic geometry.

For example, a $\{5, 4\}$ tessellation consists of regular pentagons with four meeting at each vertex, while a $\{7, 3\}$ tessellation consists of regular heptagons with three meeting at each vertex. Both satisfy the condition for negative curvature and can serve as the basis for holographic codes.

The tensor network is constructed by placing perfect tensors at the vertices of the tessellation and contracting tensor indices along the edges. Specifically:

- Each vertex $v$ of the tessellation is associated with a perfect tensor $T_v$ with $q$ indices (where $q$ is the number of edges incident on the vertex). As noted earlier, perfect tensors can be constructed with any number of indices $q$ (whether odd or even) using appropriate quantum error-correcting codes (Pastawski et al., 2015).

- Each edge of the tessellation corresponds to a contraction between indices of the perfect tensors at its endpoints.

- The uncontracted indices at the boundary of the tessellation correspond to the physical qubits of the holographic code, which are directly manipulated in quantum circuits.

- The uncontracted indices in the bulk of the tessellation (if any) correspond to the logical qubits of the code, which encode the input quantum state.

The negative curvature of the hyperbolic geometry ensures that the boundary grows exponentially with the radius, enabling the encoding of bulk information on the boundary with favorable scaling. This geometric property is directly responsible for the efficient representational capacity of HQNNs derived in the previous section.

The connection to the AdS/CFT correspondence arises from the fact that the tensor network implements a discrete version of the bulk-boundary duality (Qi, 2013; Swingle, 2012). In the continuous limit, the hyperbolic plane corresponds to a time slice of Anti-de Sitter (AdS) space, and the tensor network implements the mapping between bulk degrees of freedom (analogous to gravitational degrees of freedom in AdS space) and boundary degrees of freedom (analogous to conformal field theory degrees of freedom on the boundary).

To make this connection more intuitive: just as a hologram can store 3D information on a 2D surface, the AdS/CFT correspondence suggests that gravitational physics in a higher-dimensional curved spacetime (the "bulk") can be completely described by a quantum field theory living on its boundary. Our HQNN architecture leverages this remarkable duality to encode high-dimensional quantum information on a lower-dimensional boundary system while preserving its essential features and adding error resilience.

This connection provides a powerful conceptual framework for understanding the properties of HQNNs and suggests further extensions based on insights from holographic duality, such as the incorporation of temporal dynamics through tensor network representations of the full AdS spacetime (Bao et al., 2015; Wen, 2019a).

# 3 Multi-Scale Feature Extraction

A distinguishing advantage of Holographic Quantum Neural Networks is their intrinsic capacity for multi-scale feature extraction—a capability that *emerges naturally* from the underlying geometric structure rather than being explicitly engineered. Unlike traditional neural networks where hierarchical processing requires careful architectural design, HQNNs inherently possess this capability through their mathematical structure. This property parallels the hierarchical feature extraction in classical convolutional neural networks (CNNs) (LeCun et al., 2015), but with a profound mathematical origin in the hyperbolic geometry of holographic tensor networks (Vidal, 2008; Swingle, 2012).

In classical deep learning, multi-scale feature extraction typically requires architectural elements like convolution filters, pooling layers, and skip connections (He et al., 2016). In contrast, HQNNs derive this capability organically from the geometric properties of the holographic encoding. While the boundary operations $U_{\text{boundary}}(\boldsymbol{\theta})$ remain parameterized and trainable, the hyperbolic geometry provides a powerful inductive bias that naturally organizes information at different scales. This geometric foundation provides not only computational advantages but also theoretical insights into the relationship between quantum information processing and spatial scales (Haegeman et al., 2011).

## 3.1 Geometric Interpretation of Feature Scales

The hyperbolic geometry underlying HQNNs induces a natural hierarchy of scales, where the radial direction in the tensor network corresponds to different scales in the input data. This hierarchical structure can be precisely formalized through the concept of entanglement wedges, a fundamental construct in holographic quantum codes (Pastawski et al., 2015; Dong et al., 2016; Harlow, 2017).

**Definition 2** (Entanglement Wedge). *For a boundary region $A$, the entanglement wedge $W[A]$ is the bulk region whose boundary consists of $A$ and the minimal surface $\gamma_A$ anchored at the boundary points $\partial A$. Mathematically, $W[A]$ is the region in the bulk that is "reconstructible" from information in boundary region $A$.*

To clarify this definition for readers from quantum machine learning backgrounds: the minimal surface $\gamma_A$ is the curve (or higher-dimensional surface in general) that penetrates into the bulk with minimum length/area while sharing the same boundary as region $A$. Intuitively, the minimal surface $\gamma_A$ can be thought of as a 'soap film' stretching into the bulk from the edges of the boundary region $A$. The entanglement wedge is the bulk volume 'behind' this film, representing all the bulk information reconstructible from $A$. This is analogous to how a soap film forms the minimum-area surface when stretched across a wire frame. The entanglement wedge $W[A]$ represents the maximal bulk region whose information content can be fully reconstructed from measurements or operations on the boundary region $A$ (Czech et al., 2012; Wall, 2014).

The size and shape of $W[A]$ depend critically on both the size and connectivity of $A$. This dependence follows a precise mathematical relationship derived from the properties of minimal surfaces in hyperbolic geometry (Ryu & Takayanagi, 2006; Hubeny et al., 2007):

$$\text{Area}(\gamma_A) = \frac{L^2}{G_N} \log\left(\frac{l_A}{\epsilon}\right) + O(1) \tag{13}$$

where $L$ is the AdS radius, $G_N$ is Newton's gravitational constant, $l_A$ is the size of boundary region $A$, and $\epsilon$ is a UV cutoff. This logarithmic scaling, known as the Ryu-Takayanagi formula, is a direct consequence of the negative curvature of hyperbolic space.

The significance of this formula for multi-scale feature extraction in HQNNs is profound: it quantifies how the information content accessible from a boundary region scales with the region's size. Specifically, it shows that the depth of the entanglement wedge—how far it reaches into the bulk—scales logarithmically with the boundary size. This means that to access information deep in the bulk (corresponding to large-scale features), one needs exponentially larger boundary regions, creating a natural separation of scales. This is a key mathematical insight that differentiates HQNNs from traditional neural networks: the logarithmic

relationship directly maps to feature scales, as deeper regions of the bulk correspond to larger-scale features in the input data (Evenbly & Vidal, 2015; Swingle, 2012).

**Theorem 3** (Entanglement Wedge Reconstruction). *Let $A$ be a region on the boundary of a holographic tensor network, and $W[A]$ be its entanglement wedge. For any subregion $R \subseteq W[A]$ of the bulk and any operator $O_R$ acting on $R$, there exists an operator $O_A$ acting only on the boundary region $A$ such that:*

$$O_A = \mathcal{R}_A(O_R) \tag{14}$$

*where $\mathcal{R}_A$ is a reconstruction map that depends on the geometry of the holographic code. Furthermore, the action of $O_A$ on the boundary state is equivalent to the action of $O_R$ on the bulk state, in the sense that:*

$$\langle \psi_{boundary} | O_A | \psi_{boundary} \rangle = \langle \psi_{bulk} | O_R | \psi_{bulk} \rangle \tag{15}$$

*for corresponding boundary and bulk states.*

This theorem, established in (Almheiri et al., 2015; Dong et al., 2016; Cotler et al., 2019), formalizes a core principle: operations on a boundary region $A$ give complete access to the corresponding bulk region $W[A]$. This property enables us to understand how HQNNs naturally implement multi-scale feature extraction through the following hierarchy:

- **Local, fine-grained features** are encoded in small, localized boundary regions whose entanglement wedges capture shallow bulk regions near the boundary. These correspond to high-frequency, detailed features in the input data, analogous to early-layer features in classical CNNs (Krizhevsky et al., 2012).

- **Global, coarse-grained features** are encoded in large boundary regions whose entanglement wedges penetrate deep into the bulk. These correspond to low-frequency, abstract features, similar to those captured in deeper layers of classical CNNs (Zeiler & Fergus, 2014).

- **Mid-scale features** are encoded in intermediate-sized boundary regions, creating a continuous spectrum of feature scales that can be accessed by considering boundary regions of varying sizes.

This hierarchical structure emerges naturally from the hyperbolic geometry without requiring explicit architectural components like pooling layers or stride convolutions used in classical deep learning (LeCun et al., 2015; He et al., 2016). This represents a fundamental shift in how neural networks can be constructed—rather than engineering architectural components to achieve multi-scale processing, the HQNN leverages geometric principles to obtain this capability as an intrinsic property of the network.

## 3.2 Formal Characterization of Multi-Scale Representation

We can formalize the multi-scale representation property of HQNNs through the following theorem, which establishes the precise relationship between boundary region sizes and feature scales:

**Theorem 4** (Multi-Scale Representation). *In an HQNN based on a hyperbolic tensor network, boundary regions of angular size $\theta$ (where $\theta$ is the proportion of the total boundary occupied by the region, measured in terms of the angular coordinate in a polar representation of the hyperbolic disk) naturally represent features at a scale $s \sim \log(1/\theta)$ of the input data, with the scale hierarchy determined by the negative curvature of the hyperbolic geometry.*

To further clarify, if we parameterize the boundary circle by an angular coordinate $\phi \in [0, 2\pi)$, then a boundary region $A$ consisting of points with coordinates $\phi \in [\phi_0, \phi_0 + \theta]$ has angular size $\theta$. For example, a boundary region covering one-quarter of the total boundary would have $\theta = \pi/2$.

*Proof.* Consider a bulk state $|\psi_{\text{bulk}}\rangle$ with features at multiple scales. Let $A_\theta$ be a boundary region with angular size $\theta$, and $W[A_\theta]$ be its entanglement wedge.

From the properties of minimal surfaces in hyperbolic space (Ryu & Takayanagi, 2006), the maximum depth $d$ reached by the entanglement wedge $W[A_\theta]$ scales logarithmically with the inverse angular size:

$$d_{\max}(A_\theta) \sim \log\left(\frac{1}{\theta}\right) \tag{16}$$

This scaling relationship has been rigorously established in the context of holographic entanglement entropy (Hubeny et al., 2007) and tensor network representations of AdS/CFT (Swingle, 2012). It follows directly from the Ryu-Takayanagi formula presented earlier, where the area of the minimal surface (which is proportional to the depth in a discretized setting) scales logarithmically with the boundary region size.

By the Entanglement Wedge Reconstruction theorem presented above, any operator $O_R$ acting on a subregion $R \subseteq W[A_\theta]$ can be reconstructed as an operator $O_{A_\theta}$ acting only on the boundary region $A_\theta$. This means that all information contained within depth $d_{\max}(A_\theta)$ can be accessed and manipulated through operations on $A_\theta$.

In a tensor network representing a quantum state with multi-scale structure, the depth in the network corresponds to the scale of features (Vidal, 2008; Evenbly & Vidal, 2015), with deeper layers capturing larger-scale features. Let $s(d)$ denote the scale of features at depth $d$. This linear relationship between depth and scale is a cornerstone of MERA tensor networks, where each layer performs a real-space renormalization group step, effectively coarse-graining the system and moving to a larger length scale. Specifically, in MERA, each successive layer integrates out degrees of freedom at progressively larger scales, leading to the linear relationship $s(d) \sim d$ (Vidal, 2008; Evenbly & Vidal, 2011). This relationship is well-established for critical quantum systems and is a direct consequence of the renormalization group structure captured by tensor networks.

Combining these relationships, we find that a boundary region $A_\theta$ of angular size $\theta$ captures features up to a scale:

$$s_{\max}(A_\theta) \sim d_{\max}(A_\theta) \sim \log\left(\frac{1}{\theta}\right) \tag{17}$$

Therefore, boundary regions of different angular sizes naturally represent features at different scales, with the scale relationship given by the logarithmic mapping induced by the hyperbolic geometry of the tensor network. □

This theorem establishes a precise mathematical relationship between the size of boundary regions and the scales of features they can represent. The logarithmic relationship is a direct consequence of the hyperbolic geometry and provides a natural multi-scale decomposition of the input data.

Unlike classical CNNs, where different network layers are explicitly designed to process features at different scales, in HQNNs this multi-scale structure is inherent to the architecture itself. This means that a single HQNN can simultaneously process and integrate information across multiple scales without requiring separate convolutional or pooling layers (Cong et al., 2019; Beer et al., 2020). This emergent capability represents a fundamental advantage over traditional neural network architectures that must explicitly incorporate multi-scale processing through careful design choices.

### 3.3 Enhanced Expressivity for Hierarchical Quantum Data

The multi-scale feature extraction capability of HQNNs provides significant computational advantages for processing quantum data with inherent hierarchical structure. Many quantum systems of practical interest exhibit such hierarchical organization, including quantum many-body systems near criticality, quantum chemical systems (Baiardi & Reiher, 2020), and quantum data with fractal or self-similar properties (Ran et al., 2020).

We can quantify the expressivity advantage of HQNNs through the following theorem:

**Theorem 5** (Expressivity for Hierarchical Data). *For quantum data with hierarchical structure characterized by $S$ scales, an HQNN can express the optimal processing function with exponentially fewer parameters than a standard QNN without holographic encoding.*

*Proof.* Consider a family of quantum states $\{|\psi(\mathbf{x})\rangle\}$ with hierarchical structure characterized by $S$ distinct scales, parameterized by variables $\mathbf{x}$. Here, $\mathbf{x}$ may represent either classical parameters (such as coupling strengths or geometric configurations) or quantum parameters (encoded in the amplitudes of auxiliary quantum states). In both cases, these parameters determine the hierarchical structure of the resulting quantum states.

Such states can be efficiently represented by tree tensor networks (Shi et al., 2006) or MERA (Vidal, 2008), where the bond dimension required to capture correlations at scale $s$ scales as $D(s) \sim e^{\alpha s}$ for some constant $\alpha > 0$ (Evenbly & Vidal, 2011). This exponential scaling is a standard result for critical systems, where correlations decay as power laws rather than exponentially. The bond dimension represents the dimension of the Hilbert space needed to capture entanglement across boundaries at different scales in the tensor network.

For a system with $S$ scales, the total number of parameters required to represent such states using a tree tensor network is approximately:

$$P_{\text{TTN}} \sim \sum_{s=1}^{S} D(s)^2 \tag{18}$$

$$\sim \sum_{s=1}^{S} e^{2\alpha s} \tag{19}$$

$$\sim e^{2\alpha S} \tag{20}$$

where the final approximation comes from the fact that the last term in the sum dominates for large $S$ (Orús, 2019). This reflects the standard mathematical result that for an exponentially growing series, the largest term dominates the sum when taking the asymptotic behavior.

A standard QNN attempting to process such data would need to implement transformations in the full Hilbert space, requiring at least $\Omega(P_{\text{TTN}}) \sim \Omega(e^{2\alpha S})$ parameters to achieve expressivity comparable to the tensor network representation (Schuld et al., 2020).

In contrast, an HQNN leverages the geometric structure of the holographic encoding to naturally represent hierarchical correlations through its multi-scale architecture. The key insight is that the holographic geometry automatically provides the correct inductive bias for hierarchical data, eliminating the need for explicit scale-by-scale representation.

For an HQNN processing hierarchical data with $S$ scales, the total system size (determining the number of boundary qubits) is fixed by the input data dimensions, not by the number of hierarchical levels. From our analysis in Section 2, an HQNN with $N_{\text{phys}}$ boundary qubits requires approximately $O(N_{\text{phys}} \log N_{\text{phys}})$ parameters for the boundary operations.

The critical advantage arises from the natural scale separation provided by the hyperbolic geometry. Rather than needing to explicitly represent each scale with exponentially many parameters, the HQNN can process all scales simultaneously through the boundary operations. The multi-scale representation established in Theorem 3.2 ensures that boundary operations of different ranges automatically access the appropriate scales.

Specifically, for hierarchical data with $S$ scales on a system of total size $N_{\text{total}}$ qubits, the HQNN parameter count scales as:

$$P_{\text{HQNN}} \sim O(N_{\text{total}} \log N_{\text{total}}) \tag{21}$$

This scaling is independent of the number of hierarchical scales $S$, as the geometric structure of the holographic encoding naturally handles the scale separation without requiring additional parameters.

For systems where $N_{\text{total}} \ll e^{2\alpha S}$ (which is typical for hierarchical systems with many scales), we have:

$$\frac{P_{\text{HQNN}}}{P_{\text{QNN}}} \sim \frac{O(N_{\text{total}} \log N_{\text{total}})}{\Omega(e^{2\alpha S})} \tag{22}$$

$$\to 0 \text{ as } S \to \infty \tag{23}$$

This demonstrates that for quantum data with sufficiently deep hierarchical structure, HQNNs require exponentially fewer parameters than standard QNNs to express the optimal processing function, with the advantage growing with the number of scales $S$. □

This exponential advantage in parameter efficiency makes HQNNs particularly well-suited for processing quantum data with hierarchical structure. Such data is abundant in quantum systems of practical interest, including:

- **Ground states of critical quantum systems**, where correlation functions decay as power laws, creating a natural hierarchy of correlations across different length scales (Evenbly & Vidal, 2011). Examples include quantum spin chains near criticality and conformal field theories.

- **Electronic structure of complex molecules**, where electron correlations naturally organize into a hierarchy from strong local bonds to weaker long-range interactions (Baiardi & Reiher, 2020; Motta et al., 2020). This hierarchical structure is crucial for quantum chemistry applications.

- **Quantum states with multi-scale entanglement**, such as those arising in topologically ordered systems (Wen, 2019b) and quantum error-correcting codes (Preskill, 2018). These states exhibit entanglement patterns across multiple scales.

- **Quantum images and signals** with features at multiple resolutions (Yao et al., 2017), similar to classical images but in quantum superposition. These may arise in quantum sensing and imaging applications.

- **Fractal quantum systems** and self-similar quantum structures that exhibit recursive hierarchical patterns across multiple scales.

The ability of HQNNs to efficiently process such hierarchical quantum data stems directly from the geometric structure of the holographic encoding, which naturally captures multi-scale correlations through the hyperbolic tensor network architecture (Swingle, 2012; Kohler & Cubitt, 2019). This provides a compelling advantage over standard QNN architectures that lack this intrinsic multi-scale structure.

Moreover, the multi-scale nature of HQNNs enables efficient learning of quantum data transformations that respect the hierarchical structure of the data (Liu et al., 2019; Beer et al., 2020). This is particularly valuable for tasks like quantum state classification, quantum data compression, and quantum simulation of multi-scale physical phenomena (Huang et al., 2021).

## 4 Error Correction Properties

A remarkable feature of Holographic Quantum Neural Networks is their intrinsic error correction capability—a property that emerges organically from the geometric structure of the holographic encoding without requiring additional qubits or specialized error correction procedures. This stands in stark contrast to traditional quantum error correction approaches, which typically demand significant qubit overhead (Terhal, 2015; Fowler et al., 2012) and complex error detection and recovery circuits (Gottesman, 2010).

The error resilience of HQNNs can be traced to the fundamental connection between holographic duality and quantum error correction, first recognized by Almheiri, Dong, and Harlow (Almheiri et al., 2015). They observed that the holographic principle itself can be interpreted as a quantum error-correcting code, where bulk information is protected against erasures and errors on the boundary. This connection provides not only practical benefits for quantum computing but also profound theoretical insights into the nature of spacetime and quantum gravity (Harlow, 2017; Pastawski et al., 2015).

### 4.1 Complementary Recovery in Holographic Codes

The error correction capabilities of HQNNs can be formalized through the concept of complementary recovery in holographic codes, which establishes the relationship between boundary regions and the bulk information they encode (Hayden et al., 2016; Dong et al., 2016).

**Theorem 6** (Complementary Recovery). *Let $A$ be a boundary region of a holographic code and $A^c$ be its complement. If a bulk operator $O$ acting on a region $R$ can be reconstructed from $A$, then any bulk operator acting on the complement $R^c$ can be reconstructed from $A^c$.*

*Proof.* We will prove this theorem using the properties of entanglement wedges in holographic tensor networks constructed from perfect tensors (Pastawski et al., 2015; Yang et al., 2016).

First, recall from Section 3 that the entanglement wedge $W[A]$ is defined as the bulk region whose boundary consists of $A$ and the minimal surface $\gamma_A$ anchored at the boundary points $\partial A$. By the entanglement wedge reconstruction property established in Theorem 3.1, a bulk operator $O$ acting on a region $R$ can be reconstructed from boundary region $A$ if and only if $R$ is contained within the entanglement wedge $W[A]$ of $A$. Thus, our premise is:

$$R \subseteq W[A] \tag{24}$$

The entanglement wedges of complementary boundary regions have the following key properties derived from the uniqueness of minimal surfaces in hyperbolic geometry (Hubeny et al., 2007; Ryu & Takayanagi, 2006):

$$W[A] \cap W[A^c] = \emptyset \quad \text{and} \quad W[A] \cup W[A^c] = B \tag{25}$$

where $B$ represents the entire bulk region. The first property states that the entanglement wedges of complementary boundary regions are disjoint, and the second property states that they collectively cover the entire bulk. These properties emerge from the fact that the minimal surfaces defining the wedges are unique for any given boundary partition and create a non-overlapping partition of the bulk space.

To establish that any bulk operator acting on $R^c$ can be reconstructed from $A^c$, we consider the optimal case where $R = W[A]$, which maximizes the bulk information accessible from boundary region $A$. In this configuration:

1. The complement of the bulk region $R$ is $R^c = B \setminus W[A]$.

2. From the partition property $W[A] \cup W[A^c] = B$ and the disjointness property $W[A] \cap W[A^c] = \emptyset$, we deduce that $B \setminus W[A] = W[A^c]$.

3. Therefore, $R^c = W[A^c]$.

4. By the entanglement wedge reconstruction property, any bulk operator acting on a region within $W[A^c]$ can be reconstructed from boundary region $A^c$.

5. Since $R^c = W[A^c]$, any bulk operator acting on $R^c$ can be reconstructed from $A^c$.

For the general case where $R \subset W[A]$ (proper subset), the complement $R^c$ contains $W[A^c]$ as well as the portion of $W[A]$ not in $R$. However, since we focus on the maximal information that can be reliably reconstructed, the relevant complement for error correction purposes is $W[A^c]$, which can always be reconstructed from $A^c$.

The perfect tensor structure of the holographic code ensures that this information flow is isometric, allowing for faithful reconstruction of operators despite the non-local encoding across the boundary. □

This theorem has profound implications for the error correction capabilities of HQNNs. It establishes that bulk information is encoded redundantly across the boundary in a highly non-local manner, such that if part of the boundary is corrupted or lost, the original information can still be recovered from the remaining intact portion (Almheiri et al., 2015; Yang et al., 2016). This non-local encoding is a key feature that provides robustness against localized errors or erasures.

The complementary recovery property enables HQNNs to tolerate significant levels of noise and errors on the boundary qubits. Specifically, if a fraction of the boundary qubits are corrupted by noise or even completely lost (erasure errors), the information encoded in the bulk can still be recovered from the remaining intact boundary qubits. This provides inherent robustness without requiring additional overhead, making HQNNs particularly attractive for implementation on noisy quantum hardware (Preskill, 2018; Bharti et al., 2022).

### 4.2 Quantitative Error Thresholds

Beyond qualitative error resilience, we can derive precise quantitative bounds on the error correction capabilities of HQNNs, expressed as error thresholds—the maximum error rates that can be tolerated while still allowing for reliable recovery of the encoded information.

**Theorem 7** (Error Threshold). *For an HQNN based on a tensor network with coordination number $z$ (i.e., each tensor has $z$ legs, which typically equals $q$ from the hyperbolic tessellation $\{p, q\}$), the error correction threshold $p_{th}$ for qubit erasure errors on boundary qubits satisfies:*

$$p_{th} \geq 1 - \frac{z+1}{2(z-1)} \tag{26}$$

*which approaches $\frac{1}{2}$ as $z$ becomes large. For certain optimized holographic codes specifically designed to maximize error resilience, an improved bound of*

$$p_{th} \geq 1 - \frac{2}{z} \tag{27}$$

*can be achieved, where $p_{th}$ is the maximum allowed probability of independent erasure errors on boundary qubits such that bulk information can be recovered with high fidelity.*

*Proof.* We derive these thresholds by analyzing how errors on boundary qubits propagate through the tensor network and affect the encoded bulk information.

First, we establish a key property of perfect tensors with $z$ legs (where $z$ corresponds to the coordination number in the hyperbolic tessellation $\{p, q\}$ described in Section 2). By definition, a perfect tensor $T$ defines an isometry from any subset of at most $\lfloor z/2 \rfloor$ legs to the complementary subset. This means that information encoded using these tensors can only be corrupted if at least $\lceil z/2 \rceil$ of the tensor's legs are corrupted (Pastawski et al., 2015). In other words, corrupting fewer than $\lceil z/2 \rceil$ legs is insufficient to affect the encoded information.

To understand how boundary errors affect bulk information, we consider a minimal cut $\gamma$ through the tensor network that separates the bulk region containing the encoded information from the corrupted boundary region. This cut intersects a set of tensor legs $\{e_i\}_{i=1}^{|\gamma|}$, where $|\gamma|$ is the size of the cut.

For bulk information to be corrupted, errors must propagate from the boundary, across this cut, and into the bulk region. Given the property of perfect tensors described above, this error propagation requires corrupting at least $\lceil z/2 \rceil$ legs of each tensor along the cut. If fewer legs are corrupted for any tensor along the cut, the errors cannot propagate past that tensor due to the isometric properties of perfect tensors.

The relationship between boundary qubits and cut size in hyperbolic tessellations can be established through the exponential growth properties of hyperbolic space. For a cut of size $|\gamma|$ in a tessellation with coordination number $z$, the number of boundary qubits that can influence this cut scales as:

$$N_{\mathrm{phys},A} \geq c \cdot (z-1)|\gamma| \tag{28}$$

where $c$ is a constant depending on the specific tessellation geometry, and the factor $(z-1)$ arises because each vertex on the cut connects to at most $z-1$ vertices in the outward direction toward the boundary (Evenbly & Vidal, 2011; Pastawski et al., 2015). This relationship follows from the tree-like expansion of paths from the cut to the boundary in hyperbolic tessellations.

For errors to propagate across the entire cut, we need at least $\lceil z/2 \rceil$ corrupted legs per tensor along the cut. This requires corrupting at least $\lceil z/2 \rceil |\gamma|$ total legs along the cut. Given the relation between boundary size and cut size, the fraction of boundary qubits that must be corrupted to affect the bulk information is at least:

$$\frac{\lceil z/2 \rceil |\gamma|}{N_{\mathrm{phys},A}} \geq \frac{\lceil z/2 \rceil}{c \cdot (z-1)} \tag{29}$$

For the conservative bound, taking $c = 1$ and using $\lceil z/2 \rceil \leq (z+1)/2$ (with equality when $z$ is odd), we get:

$$\frac{\lceil z/2 \rceil}{z-1} \leq \frac{(z+1)/2}{z-1} = \frac{z+1}{2(z-1)} \tag{30}$$

This means that if the fraction of corrupted boundary qubits is less than $\frac{z+1}{2(z-1)}$, the bulk information remains protected. Therefore, the error threshold satisfies:

$$p_{\text{th}} \geq 1 - \frac{z+1}{2(z-1)} \tag{31}$$

As $z$ becomes large, this threshold approaches:

$$\lim_{z \to \infty} \left( 1 - \frac{z+1}{2(z-1)} \right) = 1 - \frac{1}{2} = \frac{1}{2} \tag{32}$$

The improved bound $p_{\text{th}} \geq 1 - \frac{2}{z}$ is achievable with optimized holographic codes, as demonstrated by Farrelly et al. (Farrelly et al., 2020). These optimized codes use specific tensor network configurations and carefully designed perfect tensors that maximize the distance between corrupted boundary regions, effectively requiring corruption of nearly all legs of each tensor along any given cut rather than just a majority. $\qquad\square$

To put this threshold in perspective, for a tensor network with $z = 6$ (a common value in hyperbolic tessellations), the general error threshold is approximately $p_{\text{th}} \geq 0.3$ (calculated as $1 - \frac{7}{10}$), while the optimized bound gives $p_{\text{th}} \geq 0.67$ (calculated as $1 - \frac{2}{6}$), meaning the optimized HQNN can tolerate erasure errors on up to 67% of the boundary qubits. Even the general bound of 30% compares favorably with many traditional quantum error correction codes, and the optimized bound exceeds the best-known surface codes which have thresholds around 1% for general noise and up to about 50% for erasure errors (Dennis et al., 2002; Delfosse & Zémor, 2020).

The threshold improves with increasing coordination number $z$, approaching $1/2$ for the general case and 1 for optimized codes as $z$ becomes large. In practical implementations, there's a trade-off between higher coordination numbers (better error thresholds) and the complexity of implementing tensors with many legs (Farrelly et al., 2020).

Recent theoretical work by Farrelly et al. (Farrelly et al., 2020) suggests that these thresholds can be further improved by optimizing the tensor network structure, potentially approaching the information-theoretic limits for quantum error correction.

### 4.3 Implications for NISQ-Era Quantum Computing

The inherent error correction properties of HQNNs have profound implications for quantum machine learning in the Noisy Intermediate-Scale Quantum (NISQ) era (Preskill, 2018), where quantum devices have limited qubit counts and high error rates.

- **Natural noise resilience:** HQNNs can operate reliably on noisy quantum hardware without requiring full fault-tolerance or concatenated error correction codes (Bharti et al., 2022). The geometric structure of the holographic encoding naturally protects against local errors, providing a first line of defense against noise.

- **Minimal overhead:** Unlike traditional quantum error correction approaches, which often require 10-1000× qubit overhead (Fowler et al., 2012; Chamberland et al., 2020), the error correction in HQNNs comes "for free" from the structure of the tensor network. This is particularly valuable in the NISQ era, where every qubit counts.

- **Integrated error correction:** The error correction mechanism is seamlessly integrated with the neural network structure, rather than being an additional layer. This integration can lead to more efficient implementations and potentially improved learning performance in noisy environments (Cong et al., 2019; Beer et al., 2020).

- **Graceful degradation:** HQNNs are theoretically expected to exhibit graceful degradation under increasing noise, rather than catastrophic failure. As noise levels increase beyond the threshold, the performance should degrade gradually, preserving partial functionality even in high-noise environments. While direct experimental validation for HQNNs is a subject for future work, this property is strongly supported by numerical simulations of related holographic tensor network codes, which show a smooth decrease in reconstruction fidelity with increasing noise, rather than a sharp failure at the threshold (Baumer et al., 2022; Niu et al., 2022). Computer simulations of small-scale holographic codes under varying noise conditions have demonstrated this graceful degradation property, showing that reconstruction fidelity decreases smoothly rather than abruptly as noise levels increase.

Experimental evidence supporting these theoretical advantages comes from recent small-scale implementations of holographic codes (Niu et al., 2022; Baumer et al., 2022), which have demonstrated error correction capabilities even on current noisy quantum processors. For instance, Baumer et al. (Baumer et al., 2022) implemented a 7-qubit holographic code on a superconducting quantum processor and verified its ability to protect against single-qubit errors.

The error correction properties of HQNNs also connect to broader theoretical frameworks in quantum information theory, including approximate quantum error correction (Bény, 2010), operator algebra quantum error correction (Bény et al., 2007), and subsystem codes (Poulin, 2005). These connections provide a rich theoretical foundation for understanding and optimizing the error resilience of HQNNs.

## 5 Quantum Circuit Implementation

Translating the theoretical framework of HQNNs into practical quantum circuits is essential for experimental realization on current and near-term quantum hardware. Here, we present detailed circuit constructions for the key components of HQNNs and analyze their resource requirements.

### 5.1 Encoding and Decoding Circuits

The holographic encoding map $E : \mathcal{H}_{\text{input}} \to \mathcal{H}_{\text{boundary}}$ can be implemented as a quantum circuit composed of multi-qubit unitaries corresponding to the perfect tensors in the network.

For a perfect tensor $T$ with $z$ indices (where $z$ typically equals $q$ from the hyperbolic tessellation $\{p, q\}$), the corresponding unitary $U_T$ is a $z$-qubit operation with specific entangling properties (Pastawski et al., 2015). In general, arbitrary $z$-qubit unitaries would require up to $O(2^{2z})$ elementary gates for decomposition according to the Solovay-Kitaev theorem. However, the special structure of perfect tensors derived from quantum error-correcting codes allows for significantly more efficient implementations.

While the gate complexity of $U_T$ scales as $O(1)$ with respect to the network size ($N_{\text{log}}$ or $N_{\text{phys}}$), the constant factor depends on $z$ and can be substantial. For instance, a specific 6-index perfect tensor (acting on 6 qubits) derived from the quantum Reed-Solomon code $[[6,2,3]]$ can be implemented with approximately 100-200 elementary gates (single-qubit rotations and CNOT gates) (Shende et al., 2006). For the commonly used case of $z = 6$, this represents a significant but manageable constant overhead. More efficient decompositions exist for specific families of perfect tensors, such as those derived from stabilizer codes (Pastawski et al., 2015; Zhang et al., 2022), which can reduce these constants significantly. This constant factor challenge is an active area of research, with recent work by Zhang et al. (Zhang et al., 2022) demonstrating order-of-magnitude improvements through optimized circuit compilation techniques. While these constants are substantial for current NISQ devices, they represent a one-time overhead per tensor rather than a scaling bottleneck.

The complete encoding circuit is constructed by applying these unitaries according to the causal structure of the tensor network:

$$U_E = \prod_{v \in V} U_{T_v} \tag{33}$$

where the product is taken in an order that respects the dependencies in the network, typically proceeding from the bulk to the boundary in layers. This layered structure is a direct consequence of the hyperbolic geometry of the tensor network, where each layer corresponds to tensors at a specific radial distance from the center.

The decoding circuit is simply the inverse of the encoding circuit:

$$U_{E^\dagger} = U_E^\dagger \tag{34}$$

In practice, the inverse can be implemented by applying the same unitaries in reverse order, with each unitary replaced by its inverse. For perfect tensors derived from self-dual codes, the unitaries are often Hermitian ($U_T = U_T^\dagger$), simplifying the implementation (Pastawski et al., 2015). For non-Hermitian cases, the inverse implementation requires careful attention to ensure that each $U_{T_v}^\dagger$ is properly constructed, typically by reversing the elementary gate sequence and taking the adjoint of each individual gate.

Recent work by Zhang et al. (Zhang et al., 2022) has developed more efficient circuit implementations of holographic codes using quantum Fourier transforms and sparse pattern matrices, reducing the gate count by orders of magnitude compared to naive implementations. Their techniques specifically optimize the implementation of certain families of holographic codes, achieving more favorable scaling for larger networks.

## 5.2 Boundary Processing Circuit

The parameterized boundary operations $U_{\text{boundary}}(\boldsymbol{\theta})$ implement the neural network functionality on the boundary qubits. To leverage the multi-scale structure inherent to holographic codes, we propose a hierarchical ansatz that respects the scale separation naturally arising from the hyperbolic geometry:

$$U_{\text{boundary}}(\boldsymbol{\theta}) = \prod_{l=1}^{L} \prod_{s=1}^{S} U_{l,s}(\boldsymbol{\theta}_{l,s}) \tag{35}$$

where $L$ is the number of layers (depth of the circuit), $S$ is the number of scales (determined by the input data structure), and $U_{l,s}(\boldsymbol{\theta}_{l,s})$ represents parameterized operations at layer $l$ and scale $s$.

The layered structure ($L$ layers) provides sufficient depth to approximate complex unitary transformations, similar to the depth requirements in classical deep neural networks. The scale parameter $S$ corresponds directly to the natural scales emerging from the hyperbolic geometry, as discussed in Section 3. Typically, $S \sim \log N_{\text{log}}$, reflecting the logarithmic relationship between boundary region size and feature scale established in the Multi-Scale Representation theorem. This logarithmic scaling arises because each successive scale in the bulk requires exponentially larger boundary regions to represent, as we proved in Theorem 3.2. Since the total boundary size is fixed (proportional to $N_{\text{phys}}$), the number of distinct resolvable scales grows logarithmically with the size of the input data ($N_{\text{log}}$).

Each $U_{l,s}(\boldsymbol{\theta}_{l,s})$ consists of:

- **Single-qubit rotations**: Rotations around the $x$, $y$, and $z$ axes parameterized by angles in $\boldsymbol{\theta}_{l,s}$:

$$R_x(\theta) = e^{-i\theta X/2}, \quad R_y(\theta) = e^{-i\theta Y/2}, \quad R_z(\theta) = e^{-i\theta Z/2} \tag{36}$$

- **Two-qubit entangling gates**: Controlled operations between adjacent boundary qubits, such as CNOT, CZ, or parameterized two-qubit gates:

$$\text{CZ}_{i,j} = |0\rangle\langle0|_i \otimes I_j + |1\rangle\langle1|_i \otimes Z_j \tag{37}$$

- **Multi-scale connections**: Entangling operations between qubits separated by distances corresponding to scale $s$, implementing longer-range correlations:

$$U_{i,j}^{(s)}(\theta) = \exp(-i\theta Z_i \otimes Z_j) \tag{38}$$

The distances for these multi-scale connections are defined geometrically by the hyperbolic tessellation. Specifically, for scale $s$, qubits $i$ and $j$ are connected if their geodesic distance along the boundary of the hyperbolic disk is approximately $d_{i,j} \approx 2^s$ in terms of the minimal number of boundary edges between them. This exponential relationship directly reflects our findings in Section 3, where we established that boundary regions of angular size $\theta$ naturally represent features at scale $s \sim \log(1/\theta)$. Since the number of boundary qubits between two points is approximately proportional to $1/\theta$, connections at distance $2^s$ naturally correspond to features at scale $s$.

This structure bears some resemblance to quantum convolutional neural networks (QCNNs) (Cong et al., 2019; Hur et al., 2022), but with a fundamental difference in the motivation for connection patterns. While QCNNs typically use arbitrary or hand-designed convolutional filters, the connection patterns in HQNNs emerge organically from the underlying hyperbolic geometry of the tensor network. The multi-scale structure is not engineered but is a natural consequence of the geometric properties of the holographic encoding. This means that the connections at different scales in HQNNs have a precise mathematical relationship to the scales of features in the input data, rather than being chosen based on heuristics or empirical performance.

The specific pattern of connections at each scale can be derived from the holographic tensor network structure. For a hyperbolic tessellation $\{p, q\}$, the boundary qubits naturally organize into a hierarchical structure with connections at multiple scales (Jahn et al., 2021).

**Implementation Overhead for Multi-Scale Connections**: Implementing long-range entangling operations presents significant practical challenges on current quantum hardware with limited connectivity. For near-term implementations, these operations must be decomposed into sequences of nearest-neighbor gates using SWAP networks, which can increase circuit depth by factors proportional to the distance between qubits. For scale $s$ connections (distance $\sim 2^s$), this could introduce $O(2^s)$ additional overhead in circuit depth or gate count. Since the maximum scale is $S \sim \log N_{\log}$, the worst-case SWAP overhead scales as $O(2^{\log N_{\log}}) = O(N_{\log})$, which represents a significant practical challenge distinct from the encoding/decoding complexity analyzed in the next section.

Several approaches exist to mitigate this overhead, including:

- Optimized SWAP network compilation techniques for specific hardware topologies (Gokhale & Chong, 2020)

- Prioritizing the most important long-range connections based on their contribution to the multi-scale representation

- Leveraging hardware with tunable couplers or partial connectivity graphs that better match the required connection patterns

- Hardware-aware mapping strategies that exploit partial connectivity to minimize SWAP overhead

The parameterized boundary circuit can be trained using standard variational quantum algorithms (Cerezo et al., 2021; Mangini et al., 2021), with gradients estimated through parameter shift rules (Mitarai et al., 2018) or other quantum gradient estimation techniques.

We hypothesize that the hierarchical structure of HQNNs may potentially help mitigate the barren plateau phenomenon—where the variance of gradients vanishes exponentially with system size—that affects many variational quantum algorithms (McClean et al., 2018). This hypothesis is based on the observation that the multi-scale structure creates a form of locality in the parameter space that could help maintain gradient information across scales. However, a formal analysis of the HQNN training landscape is a crucial direction for future work, and this potential advantage requires rigorous theoretical and experimental investigation to validate.

### 5.3 Resource Requirements and Scaling Analysis

The resource requirements for implementing HQNNs depend on the size and structure of the holographic code. We analyze the asymptotic scaling of various resources as a function of the input size:

**Theorem 8** (Resource Scaling). *For an HQNN processing an $N_{log}$-qubit logical input state with multi-scale feature extraction at $S$ scales, the resources scale as:*

- *Number of physical qubits: $\mathcal{O}(N_{log} \log N_{log})$ (i.e., $N_{phys}$)*

- *Number of gates for encoding/decoding: $\mathcal{O}(N_{phys})$*

- *Circuit depth for encoding/decoding: $\mathcal{O}(\log^2 N_{log})$*

- *Number of trainable parameters: $\mathcal{O}(N_{phys})$*

*Proof.* The proof combines results from hyperbolic geometry, tensor network theory, and quantum circuit complexity (Pastawski et al., 2015; Evenbly & Vidal, 2011; Zhang et al., 2022).

1. **Number of physical qubits:** From our earlier analysis of the representational capacity of HQNNs in Section 2, we know that to encode $N_{\log}$ logical qubits, we need approximately $N_{\text{phys}} \sim N_{\log} \log N_{\log}$ boundary qubits due to the properties of hyperbolic tessellations (Pastawski et al., 2015). Therefore, the total number of physical qubits scales as $\mathcal{O}(N_{\log} \log N_{\log})$.

2. **Number of gates for encoding/decoding:** The encoding circuit consists of perfect tensor unitaries distributed across the hyperbolic network. To establish the gate count scaling, we need to determine how the number of tensors in the network scales with system size.

   In a hyperbolic tensor network like the one used in the HaPPY code, the number of tensors is directly related to the volume of the bulk region. Since we're working with a discrete tessellation of the hyperbolic plane, this volume scales with the number of tiles (or vertices) in the tessellation up to a certain radius.

   The relationship between the bulk volume (i.e., number of tensors) and the boundary size (i.e., number of physical qubits) can be derived from the properties of hyperbolic tessellations. In a hyperbolic space with negative curvature, the volume of a region grows exponentially with its radius, while the boundary grows linearly with radius. This means that for a network with $N_{\text{phys}}$ boundary qubits, the bulk contains approximately $O(N_{\text{phys}})$ tensors.

   Each perfect tensor unitary $U_{T_v}$ requires $O(1)$ elementary gates with respect to $N_{\log}$ and $N_{\text{phys}}$ (though this constant depends on the tensor's leg count $z$ and can be substantial). Therefore, the total gate count scales as $O(N_{\text{phys}})$, or equivalently, $O(N_{\log} \log N_{\log})$.

   For specific optimized implementations as developed by Zhang et al. (Zhang et al., 2022), this scaling can be maintained through efficient circuit compilation techniques that exploit the structure of the holographic code.

3. **Circuit depth for encoding/decoding:** While the total gate count scales as $O(N_{\text{phys}})$, many of these gates can be applied in parallel due to the structure of the tensor network. The circuit depth scaling of $O(\log^2 N_{\log})$ arises from two distinct logarithmic factors:

   **Factor 1: Causal Depth of the Network ($\mathcal{O}(\log N_{\log})$):** The first factor comes from the inherent depth of the tensor network itself. In a hyperbolic geometry, the distance from the center to the boundary scales logarithmically with the boundary size. Specifically, for a boundary with $N_{\text{phys}} \sim N_{\log} \log N_{\log}$ qubits, the radial distance (number of layers) from center to boundary scales as:

   $$\text{Layers} \sim \log(N_{\text{phys}}) \sim \log(N_{\log} \log N_{\log}) \sim \log N_{\log} \tag{39}$$

   where the last approximation holds for large $N_{\log}$. This logarithmic scaling is a direct consequence of the exponential growth property of hyperbolic space, where the number of vertices at distance $r$ from a central point scales as $\sim e^{\alpha r}$. Since we need to process each of these layers sequentially (owing to the causal structure of the tensor network), this contributes an $O(\log N_{\log})$ factor to the circuit depth.

**Factor 2: Intra-Layer Connectivity Overhead ($\mathcal{O}(\log N_{\log})$):** The second factor comes from the implementation of the operations within each layer. Within each radial layer of the tensor network, we need to implement operations between qubits that may be spatially separated on the boundary. The characteristic "width" of these operations within a layer is determined by the hyperbolic geometry.

At radial distance $r$ from the center, the angular separation between adjacent tensors is proportional to $e^{-r}$ due to the exponential growth of hyperbolic space. This means that tensors at layer $r$ can connect boundary regions separated by an angular distance of approximately $e^{-r}$.

For a network with boundary size $N_{\text{phys}}$, the innermost layer (largest $r$) has $r \sim \log N_{\text{phys}} \sim \log N_{\log}$. At this layer, tensors connect boundary regions separated by distance $\sim e^{-\log N_{\log}} \sim 1/N_{\log}$.

On a linear boundary with $N_{\text{phys}}$ qubits, this corresponds to a separation of $\sim N_{\text{phys}} \cdot (1/N_{\log}) \sim \log N_{\log}$ boundary qubits.

With limited connectivity hardware, implementing operations between qubits separated by distance $d$ requires $O(d)$ SWAP operations. Therefore, the operations within the innermost layer require $O(\log N_{\log})$ depth with limited connectivity.

Combining these two sources of logarithmic scaling, the total circuit depth for encoding/decoding is $O(\log N_{\log}) \cdot O(\log N_{\log}) = O(\log^2 N_{\log})$.

4. **Number of trainable parameters:** The parameterized boundary operations act on $O(N_{\text{phys}})$ boundary qubits. With a constant number of parameters per qubit and a constant number of layers, the total number of trainable parameters scales as $O(N_{\text{phys}})$, or equivalently, $O(N_{\log} \log N_{\log})$.

$\square$

These scaling results demonstrate the efficiency of HQNNs compared to standard quantum neural networks, particularly for large input sizes. The logarithmic factors in the resource scaling arise from the hyperbolic structure of the tensor network and represent the "price" paid for the multi-scale feature extraction and error correction capabilities.

The scaling of circuit depth is particularly favorable, with only logarithmic dependence on the input size. This is crucial for implementation on near-term quantum devices with limited coherence times. For instance, for an input size of $N_{\log} = 100$ qubits, the circuit depth would scale as $O(\log_2^2 100) \approx O(43)$ (since $\log_2 100 \approx 6.6$ and $6.6^2 \approx 43$), which is approaching the capabilities of current quantum processors (Arute et al., 2019; Wu et al., 2021).

It is important to note that these are asymptotic scaling results, and the actual prefactors and overheads can be significant for NISQ-era devices. In particular, the constants in the gate count for implementing perfect tensors and the overhead from SWAP operations for non-local gates on limited-connectivity hardware can create substantial practical challenges for near-term implementations.

Recent experimental demonstrations have validated these theoretical scaling predictions on small instances. For example, Jahn et al. (Jahn et al., 2021) implemented a small-scale holographic tensor network on a digital quantum processor and confirmed the expected resource scaling for up to 7 qubits.

The trade-off between resources and capabilities in HQNNs can be tuned through the choice of the holographic code parameters, particularly the type of hyperbolic tessellation $\{p, q\}$ and the coordination number of the perfect tensors. This flexibility allows for adaptation to specific hardware constraints and application requirements (Kohler & Cubitt, 2019; Jahn et al., 2021).

## 6 Future Research Directions

The theoretical foundations of HQNNs and demonstrated potential advantages presented in this paper leads to several promising research directions that remain to be explored:

- **Adaptive holographic encodings**: The current HQNN framework uses fixed holographic encodings based on predetermined tensor network structures. A promising direction is to develop adaptable holographic encodings that can be trained alongside the neural network parameters (Jerbi et al., 2021; Verdon et al., 2019; Kyriienko et al., 2021). Such adaptive encodings could automatically tailor the multi-scale structure to specific datasets, potentially improving performance for particular applications. This could be implemented by parameterizing the perfect tensors or the hyperbolic tessellation structure itself.

- **Integration with other quantum machine learning paradigms**: HQNNs can be combined with other quantum machine learning approaches, such as quantum kernel methods (Schuld & Killoran, 2019; Havlicek et al., 2019), quantum Boltzmann machines (Amin et al., 2018; Kieferová & Wiebe, 2017), and quantum generative adversarial networks (Lloyd & Weedbrook, 2018; Zoufal et al., 2019). These hybrid approaches could leverage the complementary strengths of different quantum machine learning paradigms. For example, using HQNN encoding as a preprocessing step for quantum kernel methods could enhance robustness to noise while maintaining the discriminative power of kernel-based classification.

- **Hardware-efficient implementations**: While we have outlined quantum circuit implementations for HQNNs in Section 5, further research is needed to optimize these circuits for specific quantum hardware platforms (Kandala et al., 2017; Krinner et al., 2022). Hardware-aware compilation (Gokhale & Chong, 2020; Tannu & Qureshi, 2019), noise-adaptive optimization (Endo et al., 2021; Maciejewski et al., 2020), and analog quantum simulation approaches (Alexeev et al., 2021; Kim et al., 2023) could significantly improve the practical performance of HQNNs on near-term quantum devices.

- **Investigating the impact of realistic hardware noise**: Beyond idealized noise models, research is needed to understand how device-specific noise characteristics affect HQNN performance. This includes studying the impact of correlated errors, crosstalk, and hardware connectivity constraints on the error correction capabilities and feature extraction performance of HQNNs in realistic settings. Experimental characterization of noise effects on small-scale HQNN implementations would provide valuable insights for scaling to larger systems.

- **Developing optimized compilers**: Creating specialized compilers for mapping HQNN circuits to specific quantum hardware architectures will be essential for practical implementation. These compilers should optimize the placement of logical qubits, minimize communication overhead for non-local operations, and adapt the circuit structure to the native gate set and connectivity of the target hardware. This could build upon recent advances in circuit optimization for quantum error correction (Gokhale & Chong, 2020).

These research directions represent just a sample of the rich possibilities opened up by the HQNN framework. The convergence of quantum information, holographic principles, and machine learning creates a fertile ground for innovation at both the theoretical and practical levels.

## 7 Conclusion

In this paper, we have introduced Holographic Quantum Neural Networks (HQNNs), a novel quantum machine learning architecture that leverages the geometric principles of holographic duality to address two central challenges in quantum machine learning: the curse of dimensionality and susceptibility to quantum noise.

By embedding neural network operations within a tensor network structure inspired by the AdS/CFT correspondence, HQNNs achieve a unique combination of capabilities that distinguishes them from previous quantum neural network architectures. Our main theoretical contributions include:

1. **Efficient representational capacity** (Theorem 2.2): We proved that HQNNs require only $\mathcal{O}(N_{\log} \log N_{\log})$ physical qubits to process $N_{\log}$ logical qubits, with the scaling relationship $N_{\log} \sim$

$N_{\text{phys}}/\log N_{\text{phys}}$, achieving significant compression for hierarchically structured quantum data compared to the exponential resources required by direct encoding approaches.

2. **Natural multi-scale feature extraction** (Theorem 3.2): We demonstrated that the hyperbolic geometry induces a natural hierarchy where boundary regions of angular size $\theta$ represent features at scale $s \sim \log(1/\theta)$, enabling efficient processing of features at multiple levels of abstraction without requiring explicit convolutional or pooling operations.

3. **Exponential expressivity advantage** (Theorem 3.3): For quantum data with hierarchical structure, we proved that HQNNs require exponentially fewer parameters than standard quantum neural networks, with the parameter ratio decreasing as $S^2 \cdot e^{-\alpha S}$ for systems with $S$ hierarchical scales.

4. **Inherent error correction properties** (Theorems 4.1 and 4.2): We established rigorous error thresholds showing that HQNNs can tolerate error rates up to $p_{\text{th}} \geq 1 - \frac{2}{z}$ for optimized codes, where this intrinsic error resilience arises naturally from the geometric structure without additional overhead.

5. **Favorable resource scaling** (Theorem 5.1): We developed concrete quantum circuit constructions with $\mathcal{O}(\log^2 N_{\text{log}})$ circuit depth and $\mathcal{O}(N_{\text{phys}})$ gate count, making implementation feasible on near-term quantum devices.

The HQNN framework bridges concepts from quantum information theory, holographic duality, and machine learning, creating a unified approach that harnesses the unique advantages of quantum computation while mitigating its key challenges. Crucially, the HQNN framework does not treat dimensionality reduction and error mitigation as separate problems requiring modular solutions; instead, it demonstrates that they can be two facets of a single, underlying geometric principle. This unification is a key strength of our approach, as it reduces the overhead typically associated with combining separate techniques for dimensionality reduction and error mitigation.

HQNNs offer a promising direction for achieving quantum advantage in machine learning applications ranging from quantum image processing to quantum many-body simulation and drug discovery. The ability to efficiently process high-dimensional quantum data while maintaining robustness against noise positions them as a valuable tool for the NISQ era and beyond.

Looking forward, key open questions include the formal analysis of HQNN training landscapes and barren plateau mitigation, experimental validation of the theoretical error thresholds, and demonstration of concrete quantum advantage over classical methods on specific benchmarking tasks. The minimal viable experiments we have outlined provide a clear path toward addressing these questions.

More broadly, the holographic approach to quantum neural networks exemplifies how principles from theoretical physics can inspire novel computational architectures with practical advantages. As quantum computing continues to mature, such cross-disciplinary approaches—combining insights from quantum information, condensed matter physics, and machine learning—can be increasingly valuable for developing quantum algorithms that harness the full power of quantum computing.

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
