# OpenReview forum: "Holographic Quantum Neural Networks"
_TMLR — Rejected by TMLR_

### Review · Reviewer_dsJh · 2025-08-22

**Summary Of Contributions:**

The paper introduces Holographic Quantum Neural Networks (HQNNs), which embed parameterized boundary unitaries within a holographic tensor-network encoding. The authors provide theoretical results on representational capacity, multi-scale feature extraction, and inherent error correction. The ambition is to combine ideas from holography, tensor networks, and quantum machine learning to obtain architectures that are efficient and noise-resilient.

**Additional Comments:**

Overall, this submission is ambitious and theoretically interesting, but it falls short of TMLR’s acceptance criteria. The HQNN architecture is entirely linear, resembling a holographic kernel method more than a nonlinear neural network, and the analogy to non-linear learning models is overstated. The paper does not include empirical evidence or numerical experiments to support its practical claims, and it lacks direct evaluation against state-of-the-art quantum machine learning methods (e.g., VQCs, QCNNs). For these reasons, I recommend rejection at this stage. With added empirical validation and clearer positioning relative to existing approaches, the work could potentially make an interesting contribution in the future.

**Audience:**

Yes

**Audience Explanation:**

A subset of TMLR’s readership working on quantum machine learning theory may find the cross-disciplinary ideas (holography, tensor networks, quantum error correction) conceptually stimulating. However, the broader ML audience will likely find the lack of empirical evaluation and the absence of clear benchmarking against established QML approaches limiting. Interest will therefore likely be very niche rather than general.

**Broader Impact Concerns:**

No immediate ethical or societal concerns are apparent. The work is theoretical and situated within quantum information/ML. The main concern is about scientific positioning: by overstating the resemblance to non-linear learning models and the practicality for NISQ hardware, the paper risks creating misleading impressions as long as the claims are not suitable validated empirically.

**Claims And Evidence:**

No

**Claims Explanation:**

While the paper provides mathematical proofs of certain properties (e.g., isometric encoding, asymptotic qubit scaling, and error thresholds), the broader claims — such as HQNNs potentially mitigating barren plateaus, being suitable for NISQ devices, or learning hierarchical features akin to CNNs — are not supported with empirical evidence. The architecture acts entirely linearly (i.e., encoding–unitary–decoding), making the analogy to nonlinear neural networks misleading. The paper does not evaluate HQNNs against existing state-of-the-art quantum ML architectures such as variational quantum circuits, quantum convolutional networks, or tensor-network QML methods. As such, the evidence is incomplete and not fully convincing.

**Requested Changes:**

- Clarify that the HQNN transformation is strictly linear, and avoid overstating the analogy to classical deep neural networks.
- Provide a fair evaluation against state-of-the-art alternatives (e.g., VQCs, QCNNs, MERA, TTN), rather than only (partially) contrasting with direct encoding.
- Add empirical evidence: at minimum small-scale simulations or toy tasks that demonstrate HQNN behavior in practice and substantiate claims about noise resilience and expressivity.
- More clearly articulate how HQNNs differ from quantum kernel methods, since the encoding can basically be interpreted as a feature map.

---

### Review · Reviewer_N4pv · 2025-08-28

**Summary Of Contributions:**

In this work the authors present a quantum machine learning framework for quantum neural networks that they claim can more efficiently and robustly operate on quantum data. The authors claim this approach, inspired by holographic encodings, has beneficial representational scaling, intrinsic multi-scale features, and error tolerance.


Pros:
- Brings together a lot of interesting ideas in a novel manner. The boundary processing approach is conceptually neat.
- Multiple scales for processing via hyperbolic geometry could help offer insights into other QML works and the inductive biases of similar (QCNN) models


Cons:
- There’s some lack of clarity about the cost/need for exponential representation. Specifically, in the intro it is said “quantum implementation of QNNs for high-dimensional data prohibitively expensive”, while that is certainly true for classical simulations, it isn’t clear this necessarily means for physical systems (since if I have N qubits I naturally have a physical system described by exponential scaling hilbert space, but that doesn’t mean I should delete my qubits)
- Some of the claims warrant further explanation to align with other understandings in literature
- There are some notational hangups
- The paper would meaningfully benefit from some empirical evidence. I hate to be the reviewer that says “where experiments?”, but in the case of something like QML, the drive for algorithm development is a fundamentally empirical question (like ML) and thus proposing a new algorithm without any consideration (even on small examples) seems suboptimal. While theory papers that contain no empirical evidence are valid and should be published, this is not a traditional quantum algorithm paper, in which asymptotic scaling proofs are sufficient, but a machine learning paper in which the constants and the real world scaling matters. I won’t say this is a requirement for publication, I just think it could help the paper a lot.

**Audience:**

Yes

**Audience Explanation:**

This work would be of interest to at least some of the TMLR community. The QML community is regularly in search of novel approaches to ansatz and an approach that bridges ideas from QEC would be welcomed.

**Broader Impact Concerns:**

Ethical implications are sufficiently addressed.

**Claims And Evidence:**

No

**Claims Explanation:**

- The 67% erasure threshold needs to be explained more, specifically in the context of work which indicates the limits on quantum capacity.
- The claims around “providing both compression” are unclear. Specifically, N log N qubits seems like redundancy (which makes sense in a QEC context).
- The claim that perfect tensors can be scaled to any q “this property can be generalized to any number of indices q”, should be clarified in the context of AME states which cannot be generalized to any q [2].
- “We hypothesize that the hierarchical structure of HQNNs may potentially help mitigate the barren plateau phenomenon” is a (empirically) very testable claim


[1] Bennett, C. H., DiVincenzo, D. P., & Smolin, J. A. (1997). Capacities of quantum erasure channels. Physical Review Letters, 78(16), 3217.

[2] Huber, F., Gühne, O., & Siewert, J. (2017). Absolutely maximally entangled states of seven qubits do not exist. Physical review letters, 118(20), 200502.

**Requested Changes:**

- Fix minor typographical errors (e.g. the quotes look off, the leading double quote and the trailing double quote look like different formats)
- The paper would benefit from better organization. The full content of the proofs doesn’t need to be in the main text, and filling out a SI/appendix is warranted. This would help convey the key points better.
- N_log/phys is not super clear notation
- Strengthen the claims (as highlighted above)

---

### Review · Reviewer_GvGb · 2025-09-11

**Summary Of Contributions:**

The authors propose the mapping for the tensor networks to the neural networks using the holographic formalism. The authors leverage the boundary encoding property of the tensor networks and the natural error correcting architecture as one of the main points of the proposed holographic neural networks. In addition the authors demonstrate that due to the tensor nature of the implementation the required physical resources in the number of qubits are reduced. Finally the holographic property of the implementation gives these networks natural ability to calculate features at multiple scales.

However there are several concerns about the overall complexity that is naturally emerging as the required tensor and holographic nature of the proposed model. First the perfect tensor implementation and holographic encoding requires a lot of computational resources. Combined with the fact that the implementation requires a constant add-ons of SWAP gates and large number of unitaries might reduce the overall advantage with respect to other models or even with respect to classical implementations.

Overall the paper presents a theoretical formalism for designing HQNN with several key points not answered.

**Audience:**

Yes

**Audience Explanation:**

The usage of the HQNN is very exciting however it is not enough explained for the machine learning understanding. For instance, while the authors claims that natural property of the HQNN is the extraction of the multi-scale features it is not directly observable how this maps back to the standard CNN models of hierarchical features extractions. The fact that the HGNN reduces the amount of required physical qubits and provides an error correction facility is very exciting and possibly has efficient implementation in the future quantum computers. However it would be intersting to comment on the fact that because this methodology was formulated in the BQE framework of machine learning if it also suffers the barren plateau problem.

**Claims And Evidence:**

Yes

**Claims Explanation:**

The paper focuses mostly on theoretical aspects of the HQNN and as such it provides all the proofs for its claims. Interesting are the concepts of the holographic encoding that allows to expand the representation from the boundary to the whole tensor network and thus simulate the multi scale features.

**Requested Changes:**

Provide a more detailed mapping from the holographic mapping to the multi-level features in a standard CNN network. In particular how does the network structure change when the filter in a CNN changes its size.

An example albeit theoretical would be very helpful to machine learning community.

A estimation of the gate complexity for the implementation and its scaling with increasing number of qubits,.

---

### Decision · Action_Editor_C3Yw · 2025-10-19

**Recommendation:** Reject

**Additional Comments:**

The reviewers consider the following changes to be necessary:
1. Add empirical validation: The authors must substantiate their practical claims (e.g., regarding noise resistance, mitigation of the barren plateau problem, suitability for NISQ devices, or the ability to learn hierarchical features) through simulations or small-scale experiments.
2. Clarify and tone down the analogy to neural networks: The reviewers emphasize that the HQNN architecture is linear. Authors should clarify this and reconsider or more precisely formulate the analogy to nonlinear classical neural networks to avoid misunderstandings.
3. Thorough comparison with existing methods: It is essential that HQNNs are fairly compared with the current state of the art in quantum machine learning. This includes variational quantum circuits (VQCs), quantum convolutional networks (QCNNs), and other tensor network-based QML approaches.
4. Clarify and substantiate theoretical claims:
o The explanation of the 67% erasure threshold must be more detailed and explained in the context of quantum capacity limits.
o Claims about data compression and the generalization of “perfect tensors” need to be formulated more clearly and precisely, especially with regard to known limitations (such as AME states).
o An estimate of gate complexity and its scaling is required.
5. More detailed explanations and examples: The authors should provide a more detailed mapping of holographic features to multi-scale feature extraction in classical CNNs. Concrete examples, even if theoretical in nature, would help the ML community better understand the concepts.
6. Improve structure and notation: The organization of the paper could be improved by moving extensive proofs to an appendix. Notation (e.g., $N_{\text{log/phys}}$) and typographical errors should also be revised.

**Audience:**

Yes

**Audience Explanation:**

The reviewers agree that the work is basically interesting. The theoretical basis and ambitious goals of the manuscript, especially the combination of holography, tensor networks, and quantum ML to solve key challenges, make it an interesting contribution in terms of content.

**Claims And Evidence:**

No

**Claims Explanation:**

The claims are considered insufficiently substantiated due to a lack of empirical validation, misleading analogies to neural networks, insufficient comparison with existing methods, and unclear or insufficiently substantiated theoretical assertions.

**Resubmission Of Major Revision:**

The authors may consider submitting a major revision at a later time.